# The Main Drivers for Sustainable Decisions in a Family Business That Impact the Company's Performance

**Adriana Cioca \*, Kassam Wehbe, Delia Popescu and Constanta Popescu**

Faculty of Economics, IOSUD, Valahia University of Targoviste, 130004 Targoviste, Romania; kassam.wehbe@gmail.com (K.W.); depopescu@yahoo.com (D.P.); tantapop@yahoo.com (C.P.)

\* Correspondence: ada.cioka@gmail.com; Tel.: +40-743-332-290

**Abstract:** The successful ways in which families have conducted their businesses decade after decade have drawn scholars' attention to what the mainstream ideas are when it comes to making sustainable decisions. This article focuses on the main drivers behind sustainable decisions made by family businesses with respect to three pillars: economic, environmental, and social. In this context, the authors' aim is to present a statistical model for forecasting companies' future revenue in the next financial year by analyzing the relationship between the main internal drivers of family businesses and their corresponding financial objectives. Additionally, the analysis of the long-term strategy and the short-term actions indicates an understanding of environmental awareness. Reaction time in investment decisions represents a challenge for the sustainable performance of family companies. Human resources with good operation management in family businesses contribute to the assurance of long-term business stability and high returns on investments. The results will contribute to the literature on economic sustainability of family businesses.

**Keywords:** family business; time; decision; process; environment; sustainable

---

## 1. Introduction

Family businesses account for a great number of private companies, and are perceived as the backbone of the world's economy and as sustainable business models [1]. The phenomena of globalization and hyper-competition have drawn scholars' attention regarding family businesses' fast decision-making processes, which affect their business performance [2]. Family businesses have always been capable of outperforming non-family companies in decision-making, of adapting to economic changes, and of creating their competitive advantage [3]. In a dynamic environment, under the constraint of time, family businesses leverage their opportunities by taking advantage of their attributes, such as shareholder structure, family legacy, the involvement of family members in the business, family governance, long-term vision, and risk-averse orientation. In addition, family businesses are agile, with an appetite for innovation and trustworthiness, which places them in a far superior position compared to non-family businesses [4]. This article aims to investigate the attributes that underlie sustainable family business decisions, which have a huge impact on companies' performance; the authors understand performance as the development of the revenues and their forecasts.

Today, reaction time is a key driver for success in general. The difference between the verbs "to have" or "to lose" makes family businesses think twice about their investment decisions and causes them to deploy a detailed management analysis according to their set-up strategy. Recent pieces of evidence suggest that family business governance and an organizational leadership approaches create long-term investment capability and a valuable perspective in a dynamic environment, which position

them as sustainable business organizations in the world [3]. Despite the natural differences between family businesses in terms of size, location, industry, and activity, there are certainly similarities that connect them and make them inimitable and recognizable [5]. The family ownership over the shares, the voting rights, and implicitly, the decision-making power, entail the family control. However, the unique family influence on the business involves a high degree of control over the organization and the assurance of the transfer of the business to the next generation [6]. Shanker and Astrachan ranked [7] the level of influence of the family over the business with the following: small level of family influence, the middle level of family influence, and massive influence. Through this action, the family proves its long-term commitment and ensures business continuity. Each family business's philosophy is unique and comes from the family itself, from the values, traditions, and beliefs that are divided throughout the business, finally comprising the company's strategy [8]. The founders of the family business face most of the dilemmas concerning time: which comes first, the family or the business? On the other hand, one of the consequent results of their business performance is also the involvement of the family in the running of the business [7,9]. The key for family members to be successfully involved in the family business comprises a good business education and the members' familiarization from earlier times with the family values that guide the company [10]. An efficient family involvement underpins qualitative management control over the family business's processes and, ultimately, the long-term business performance [11]. At the level of Europe, family businesses account for approximately 50% of the GDP, and their performance is one of the main key economic indicators of sustainable businesses [12].

To be able to forecast the future company revenue in the next financial year and to accomplish the objective of the paper, the authors deployed a descriptive analysis and an econometric model to the top 100 family companies of the world. In the first phase, the authors described the correlation of the main non-financial drivers behind sustainable decision-making processes in comparison with their economic correspondents for companies' performance. In the second phase, the authors empirically analyzed and tested the results of the correlations by analyzing the descriptive statistics, the correlation matrix, and the regression output, as well as by testing for the homoscedasticity of errors in regression. The authors present the study's results as well as the weaknesses and create a framework for future research on practical family business implications. Our study results will contribute to the family business literature by showing the important similitudes of family businesses as well as their financial approaches to making sustainable decisions that impact their future sustainable performance.

## 2. Literature Review

In general, there is a misconception that family businesses seem to be too small and unimportant, but they are very large, with extensive experience on the market, more stability, business heritage from generation to generation, and real economic contributions, making up approximately 50% of GDP at the level of Europe. For example, 88% of the companies in Switzerland are under the ownership of families; at the global level, 80–90% of companies are family businesses [12].

Family businesses are recognized for their fast and sustainable decisions in a dynamic environment, as well as the way that families' influences are projected onto their businesses' organizational identities; these have drawn scholars' attention for further investigation [13]. The originality of a family business is offered by the family that runs the business and its unicity comes from family power of influence, "strategy fit" [12].

The key performance indicators are the family passion for the business, family governance policy, powerful emotional attachment of the family to the company, willingness to transfer the business to the future generation, innovative tools to continuously perform the family business brand, family experience, family influence, and recognition [1,14]. A family business is characterized not necessarily by its size or by the percentage of stake owned, but most importantly, by the voting rights a family has over the business. Additionally, the degree of family control and influence over the business represents the mainstream of what makes the family business unique. The degree of family control

(as shown in Figure 1) dictates the power of family influence over the business; "the dimension of power is a direct tool of family control over business" [7,11,15]. The recent literature reveals a positive association between family control and the company's performance: e.g., Barontini and Caprio (2006) [16] analyzed 675 family business from 11 countries, over 10 years, at the level of Europe. Sacristan-Navarro (2011) [17] found over 6 years in Spain that the family involvement is much more important than the ownership structure; Garcia-Ramos (2011) [18] have concluded that the role of the family in the business influences the performance of the company.

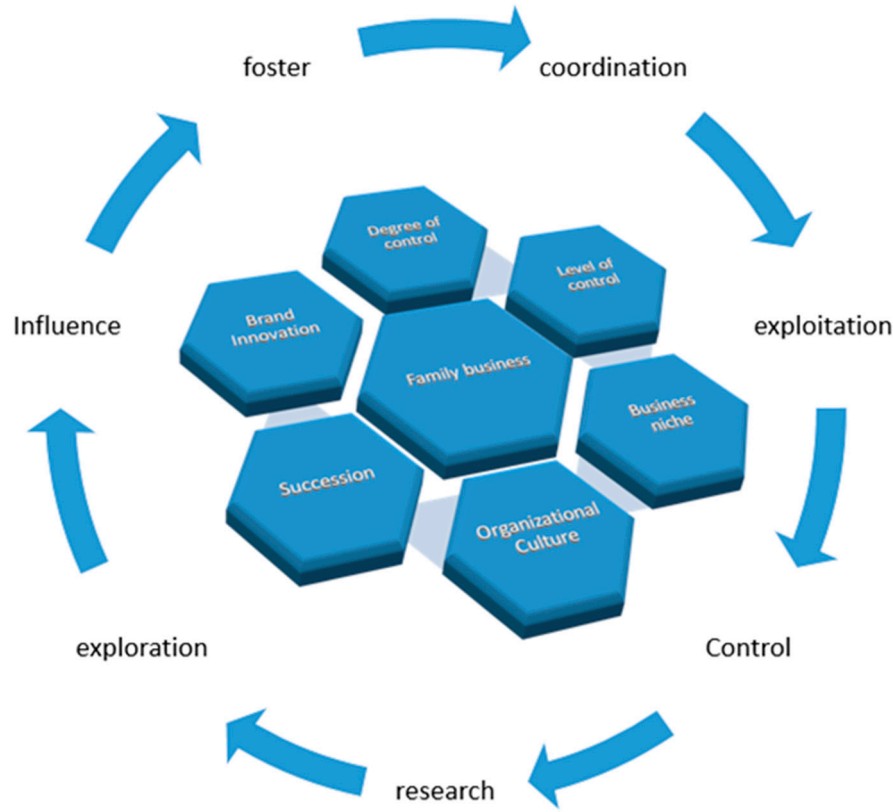

**Figure 1.** Family control degree over the business. Source: the authors.

The association between family involvement and the impact on a company's performance has been projected over time, in many articles (Garcia-Castro (2014); Chang (2009); Rutherford (2008), Mazzi (2011)) [19–22] and is defined as the sum of four components: ownership structure, governance policy, management, and succession. The time horizon of the family business's existence limits the family history within the business. The family history in business means the longevity of the business over generations. Controlling the family philosophy improves the business decision-making process, which relates to emotional attachment in the company. Moreover, communication style and respect are directly related to the control of philosophy. According to Zahra et al. (2008) [23], the family business has much more structured, flexible and higher levels of trust. According to Zellweger (2017) [11], culture reveals another emotional aspect, the difference in identity between family and business refers to family members, company reputation, and public image.

The involvement of the family members cannot be efficient without a proper business education and open communication within the family. The education of the family members underpins their future management roles in the leadership positions. Understanding the goals and organizational culture of the family business, acting by following the family philosophy and collaborating with family partners, represents an important challenge for the members to foster, to grow, and to successfully diversify the family business [24]. Moreover, the early involvement of family members in the family business will later contribute to an uninterrupted transfer in the succession process. The family

involvement in the business influences the long-term strategy and the performance viability [25]. The better the family knows the needs of its business, as easier and faster the decision-making process is.

With the continuous involvement of the family, such as ownership control and corporate governance, the business organization receives over time the patent of the family (organizational management culture), which consists of values, attitudes [9,26,27], and a set of non-economic and economic goals [11]. For family businesses, the harmony within the family and the family members' identification with the business matter in their decision-making process [28]. The uniqueness of each family member in the business strengthens the foundation of the business through beliefs, principles, trust [29], commitments [30], visions, and traditions. From the time horizon, the family model (uniqueness) embraces the business that comes with a set of long-term economic objectives: the growth of the business, preserving family capital, and company performance [9,28]. However, the family cannot run a business if the business does not fit the family pattern; this is possible in the short and medium term; otherwise, the business will be planned for sale and not transferred to future generations, as shown in Figure 2, "Organizational management culture in the family business".

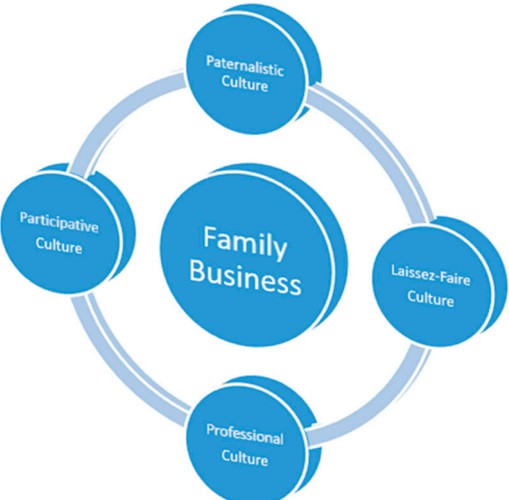

**Figure 2.** Organizational management culture in the family business. Source: the authors.

Concerning the decision-making process, Dyer [31] indicated four types of family culture characteristics that affect the company's decisions: (1) paternalistic culture—a fusion of the past values and present trends. The challenge is defined by the know-how and capacity of controlling and making fast decisions by the family; (2) laissez-faire culture—employees are empowered to make decisions by themselves; (3) participative culture—a mixture of paternalistic and laissez-faire; the family who owns the business allows employees to express their ideas and to develop their capabilities; (4) professional culture—involvement of professional managers in the family business, result-oriented strategy.

Not only the organizational culture of family business represents how decisions are performed, but also the industry in which it operates represents an indicator of the frequency of decisions that influences business sustainability and implicitly influences their performance. Time constraints in the decision-making process turns out to be a true challenge for family business performance [26]. There are industries in which family businesses activate and in which competition is at a high level, forcing family companies to make various strategic decisions in a relatively short time. (e.g., industrial production). Environmental awareness is continually changing, and it is perceived as a factor of influence over the company's performance [32]. The status defined today is hyper-competition, a challenge for the family companies to accelerate their time reaction driven by their expertise [33]. This aspect is possibly a reaction of their organizational culture in the speed of the decision-making process that leads to their performance [34,35]. Two main characteristics have been identified: dynamism and hostility [36,37]. In a dynamic environment, time reaction in investment decisions for family companies

is perceived as an unstable environment, characterized by a big load of uncertainty, increased degree of risk and stress, which makes predictions more difficult [38]. The hostile environment imposes a defined discipline on the family business to avoid the status defined as a misfortune, limited company resource consumption, or reduced financial capacity [39]. To accelerate business expansion and profit growth, family businesses always choose to have an organic investment plan [40], reinvesting profits in business expansion or the innovation process and are less exposed to financial risk, through massive lending, which would lead to a loss of independence and harsh control of the business by the family. The catalysts of the family businesses that underlie the investments represent the average of the past accumulated financial indicators, such as long-term savings, innovations financed by organic cash flows, and healthy low external financing (Research Institute, Switzerland 2018). In this way, family businesses have the competitive advantage of focusing more on long-term quality growth than other, non-family businesses.

There is a similarity in all family businesses regarding the goal of growth and long-term financial use of investments that increase the company's profitability. According to the study performed by Deloitte [28], 65% of the interviewed family businesses were oriented rather to long-term values through business leaders than to short-term financial returns; such a prioritization was projected in their daily decisions. Family business strategic planning is oriented to business growth in the long-term for sustainable performance. In their strategic endeavours, the family business intends to analyze very carefully all investments and the risk associated, as they are family capital preservation oriented and their actions need short-term drive [4]. The "zoom in/zoom out" [28] approach allows the family businesses to envision their future on two time horizons: 10–20 years and 6–12 months. The number of strategic business actions in their decision-making process in the short term will create business opportunities for the long-term results. Strategic planning may not be successfully performed without having clear governance and open communication within the family. In this respect, family businesses intended to be risk averse, especially applicable for the passive business industries, where the competition is not at a height level (e.g., tobacco industry). However, there are also other businesses, which activate in the industries with a very large hyperactive competition, where the decision-making process must be fast and efficient. In this context, "keeping up with the competition", the family business learns to change its dynamism and to learn to have a more agile behavior. Moreover, a key driver of business performance is innovation, an important factor in growing the business; consequently, family businesses are more risk tolerant than risk averse [28]. The family members need to have an open communication oriented towards family culture, goals, solving the possible opinion conflicts, restoring the family business harmony, and making final decisions which are of paramount importance for the business future.

## 3. Materials and Methods

### 3.1. The Model

After a thorough review of the literature, we have developed the following research questions (Figure 3, Research Design).

RQ1: Which are the main successful catalyst drivers for the revenue and turnover of top 100 family businesses in the world?

RQ2: How to measure non-financial characteristics?

To answer these questions, first, we selected and drew out of our paper the main important drivers of the family business, seen in the mirror with their correspondent financial-economic objective, summarized in Table 1.

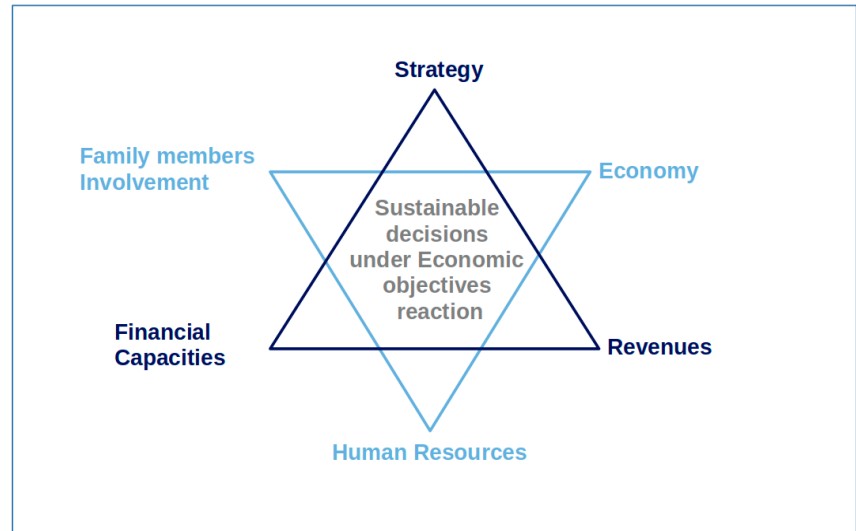

**Figure 3.** Research design. Source: the authors.

**Table 1.** Non-financial family drivers versus corresponding financial objectives.

| Family Drivers | Financial Objectives |
|---|---|
| Family control and influence | % of the family voting rights |
| Family involvement and types of family culture | % of the shares over the business |
| Harmony within the family and open communication | Long-term performance of the company |
| Expertise and discipline | Business environment awareness |
| Family business over generations | Capital preservation |
| Ability to innovate | Profit reinvested |

Source: authors.

Second, we used qualitative and quantitative research consisting of two stages. Within the first stage, we analyzed several reports regarding the situation of top-ranked family businesses, intending to identify the familys' key financial catalyst drivers, such as preferred organization, their preoccupations for value creation, acquisition, and others. Even the employees represent an important cost position for the company, and we consider them as added values for the companies because each employee contributes to the performance of the company by their experience and the fulfilment of a given task received from their managers in the company. The more a family company has experience and the more well-educated employees it has, the more additional projects it can take to grow the revenues. In this context, employees are considered an important asset for the company, they are considered cost-time efficient in their process activities and have a strong connection with the company's growing revenues.

*3.2. The Sample*

Our study focused on the top 100 family businesses in the world, ranked by the Family Capital report according to the period 2017–2019 [2,3]. We chose the top 100 family businesses in the world as to show their similarities, particularities, their economic projection with respect to the revenues, and the way they performed sustainable decisions over the years.

We analyzed the surveys issued by three consulting companies from the Big Four group which have specialized departments for family business consulting and conduct annual research on this issue, covering the period 2017–2018. Ernst & Young published "How do we create value and build trust in this transformative age?" [41] and PricewaterhouseCoopers (PwC) carried out the Global Family Business Survey 2017/2018 [42]. The survey published by KPMG, which we used in our study, refers to family businesses within the European Union (EU) [40,43] and Deloitte published the report "Connect

for impact" [4]. The surveys mentioned above include statistics for family businesses which are used in our equation. We chose the years 2017–2018 because of the electronic databases accessible online for the public. We focused on the top 100 family businesses' website data to create a database to forecast next year's revenue thanks to the turnover, the market capitalization, the market value, the number of employees, and the family shareholding percentage.

*3.3. Procedure*

We analyzed and considered companies with adequate organization for their business, skilled experts and experienced first-line managers and assumed that these companies may generate the maximum profits out of the revenue forecast. In this article, we talk about the revenues and the benefits that are made before any cost of goods sold (COGS) as the performance measure. All the characteristics are interconnected and each one of them affects all the financial processes. We gathered the characteristics into two groups, financial and non-financial characters. To compute the non-financial values, we considered family members as the family percentage shareholding, because when a family member owns more shares and voting rights of the business, they influence strategic decisions which affect the decision-making process and thus everything else [44]. Then to calculate the competition and the business environment awareness, we considered the market value, as it shows the real potential of the company. Afterwards, we calculated the business size by the number of employees [45]. We gathered the strategy and the revenue together because good revenue is the result of a good strategy. To calculate the change in the revenue, we compared it with last year's data to know how well the company evolved.

*3.4. Measures*

This report intends to present a model for forecasting future revenues based on data from the previous period. This study adopted a multiple regression model to predict future revenue values based on five independent variables. We began by exploring the variable relationships, then moved on to build and interpret the regression output. The authors assumed that next year's revenues depend on the following variables of the current year:

(1) Number of employees: Actual number of all blue- and white-colllar workers, the whole management, and all trainees the apprentices.
(2) Family shareholding: Actual % of the ownership of the family in the total shares of the company.
(3) Market capitalization: The value of all company shares quoted on the stock exchange (actual share price multiplied by the number of shares) or the price per share defined by the owners multiplied by the number of shares.
(4) Market value: Defined as a multiple of the earnings before taxes, depreciation, and amortization (EBITDA), where the multiplication depends on the industry and on deals done in the past in this industry.
(5) Revenues of the current year.

Thus, the authors adopted an additive multiple regression equation to explore how the above explanatory variables related to the response variable. Starting from the conclusions drawn in the first phase, in the second one, the following formula was adopted:

$$Revenue_{i+1} = \beta_0 + \beta_1 Employees_i + \beta_2 Family\ Shareholding_i + \beta_3 Market\ Capitalization_i + \beta_4 Market\ Value_i + \beta_5 Turnover_i + \varepsilon_i \tag{1}$$

where $\beta_0, \beta_1, \beta_2, \beta_3, \beta_4, \beta_5$ represent coefficients to be estimated using the OLS method and

$$\varepsilon_i \sim iiN\ (0, \delta^2)$$

The authors' model had as the independent variables: the family shareholding, the turnover, the market capitalization, the market value, and the number of employees, and we also included six control variables ($\beta_0$, $\beta_1$, $\beta_2$, $\beta_3$, $\beta_4$, $\beta_5$) to reduce the risk of biases and control for reliability.

The next year's revenue represents the independent variable and it can be measured by financial and non-financial indicators. Regarding the empirical results, the authors analyzed other studies [46,47] with respect to forecasting the revenues, and the most used indicators were return on equity (ROE), return on assets (ROA), and profit margin (PM), which are based on the company's annual earnings. However, the authors also assumed the following indicators to be relevant for analyzing how to use assets and to grow revenues: the turnover, the market capitalization, the family shareholding, the market value, and the number of employees, as each one of them affects the company's profitability and helps to interpret next year's revenue [15,48,49]. Besides the financial reports, the companies can write, or they are required to write, non-financial reports, which helps to complete the image of the company. In this study, the authors translated the non-financial indicators that were chosen to represent the variables for which data were available for all companies in the sample [50,51].

Regarding the independent variables, the scale of measurement represents the environment, strategy, and management. For the variable strategy, there were no clear data to measure it the same way as the other two variables; the authors analyzed what affects the strategy, the environment, and the management the most afterwards [12,52–55].

As well as control, another variable included in the model was the industry, which affects the financial performance [56,57]. All the indicators used in the model are presented in Table 2.

**Table 2.** Specifications of the variables in the model.

| Variable | Code | Description of Dependent Variable |
|---|---|---|
| Strategy and revenue | R | Revenue |
| Business size | E | Number of employees |
| Family members | F | Percentage of family shareholding |
| Financial capabilities | C | Market capitalization |
| Environment | V | Market value |

Source: authors.

### 3.5. Data and Analysis

Before fitting our regression model, it is important to first examine our variables. We do this by running summary statistics. The data are obtained from six variables of interest, each containing 70 observations. There are no missing values reported.

Table 3 below illustrates the mean, median, standard deviation, and range of our variables. It is established that 2018 revenue values vary from USD 18,404 M to USD 495,012 M with a median of USD 4700 M. The median revenue value falls below the average revenue of USD 78,910 M. This shows that the revenue data are skewed to the right.

Additionally, the number of employees included in our study ranges from 9139 to 2.3 M with a median of 133,475, while family shareholding varies from 33.4 to 99, with a median and mean value of 50.7 and 56.7, respectively.

Market capitalization ranges from USD 3421 M to USD 502,599 M, with a median and mean value of USD 39,280 and USD 75,899 M. The analysis also establishes that market value varies from 762.7 to 655,000, with an average value of 93,133.

Finally, the turnover averages at 5.4 with a minimum and maximum value of −19.23 to 42.2.

All variables are skewed to the right because their median value falls below the mean value. However, this does not violate the normality assumption since the sample size adopted is large.

To further understand our variables, a correlation analysis is done to determine the nature and strength of the relationship that exists between the dependent and independent variables.

Correlation coefficients range from −1 to +1. Coefficient values closer to 1 indicate a strong positive correlation while negative values indicate an inverse relationship. A correlation coefficient closer to zero indicates weak or no correlation.

**Table 3.** Descriptive summary statistics.

| | 2018 Revenues in M USD | 2017 Number of Employees | Family Shareholding 2017 | Market Capitalization in M USD as of 31 December 2017 | Market Value 2017 | Turnover 2017 |
|---|---|---|---|---|---|---|
| Mean | 78,910.30 | 216,165.9 | 56.70909 | 75,899.50 | 93,133.81 | 5.432727 |
| Median | 47,800.27 | 133,475.0 | 50.70000 | 39,280.50 | 24,470.00 | 3.800000 |
| Maximum | 495,012.0 | 2,300,000 | 99.00000 | 502,599.6 | 655,000.0 | 42.20000 |
| Minimum | 18,404.18 | 9139.000 | 33.40000 | 3421.600 | 762.7000 | −19.23 |
| Std. Dev. | 96,615.97 | 392,747.9 | 18.94029 | 98,710.02 | 151,614.1 | 11.35852 |
| Observations | 70 | 70 | 70 | 70 | 70 | 70 |

Source: authors.

Table 4 below records the correlation matrix for all our variables included in our model.

**Table 4.** Correlation matrix and the significance of correlation coefficients.

| Probability | 2018 Revenues in M USD | 2017 Number of Employees | Family Shareholding | Market Capitalization in M USD as of 31 December 2017 | Market Value | Turnover |
|---|---|---|---|---|---|---|
| 2018 Revenues In M USD | 1 | | | | | |
| 2017 Number Of | 0.902903 | 1 | | | | |
| Employees | 11.69525 | —— | | | | |
| | 0.000000 | —— | | | | |
| Family Shareholding | −0.202448 | −0.119051 | 1 | | | |
| | −1.151017 | −0.667598 | —— | | | |
| | 0.2585 | 0.5093 | —— | | | |
| Market Capitalization | 0.547736 | 0.431824 | −0.230406 | 1 | | |
| M USD as of December 31 2017 | 3.645088 | 2.665642 | −1.318319 | —— | | |
| | 0.001 | 0.0121 | 0.1971 | —— | | |
| Market Value | 0.731758 | 0.47342 | −0.223761 | 0.486492 | 1 | |
| | 5.977797 | 2.992487 | −1.278261 | 3.100284 | —— | |
| | 0.00000 | 0.0054 | 0.2106 | 0.0041 | —— | |
| Turnover | 0.227736 | 0.174502 | −0.098913 | 0.255434 | 0.24248 | 1 |
| | 1.302199 | 0.986724 | −0.553439 | 1.470993 | 1.39159 | —— |
| | 0.2024 | 0.3314 | 0.5839 | 0.1514 | 0.1740 | —— |

Source: authors.

From Table 4, it is important to note that the correlation between the "revenues in M USD" (dependent) variable and the "family shareholding" variable is rather the small (0.2). This depicts a weak relationship. The correlation matrix shows the relationship between the variables, and some variables relate positive relationships, and some have negative relationships with each other. The 2017 number of employees has a positive relationship with 2018 revenue in M USD. Family shareholding has a negative relationship with 2018 revenue in M USD, as well as with the 2017 number of employees. Market capitalization in M USD as of December 31 2017 is positive with 2018 revenue in M USD, the 2017 number of employees, and with market capitalization in M USD as of 31 December 2017 but, at the same time, it has a negative relationship with the family shareholding. Market value relates positively with the 2018 revenue in M USD and the 2017 number of employees, as well as with market capitalization in M USD as of 31 December 2017 but has a negative relationship with

family shareholding. Turnover related positively with 2018 revenue in M USD, the 2017 number of employees, and market capitalization in M USD as of 31 December 2017, as well as with market value, but it has a negative relationship with family shareholding. The most important explanatory variables seem to be "number of employees" and "market value" with positive correlation values of 0.9029 and 0.73, respectively.

Market capitalization presents a rather weak correlation of 0.54.

## 4. Results and Discussion

### 4.1. Coefficients Analyzed

After examining our variables, we move to the next step of testing the regression model. The authors' model assumes a linear additive model as described in Equation (1). The most important explanatory variables are "number of employees" and "market value" with positive correlation values of 0.9029 and 0.73, respectively. Table 5 below illustrates the results of the model.

**Table 5.** Regression output.

| Dependent Variable: 2018 Revenues in M USD | | | | | |
|---|---|---|---|---|---|
| **Method: Least Squares** | | | | | |
| **Included Observations: 70** | | | | | |
| **Variable** | **Coefficient** | **Std. Error** | *t*-**Statistic** | **Prob** | |
| 2017 Number Of Employees | 0.172353 | 0.013852 | 12.44224 | 0.0000 | Sig |
| Family Shareholding | −117.833 | 253.1022 | −0.46555 | 0.6453 | |
| Market Capitalization in M USD as of 31 December 2017 | 0.061455 | 0.056705 | 1.083758 | 0.2881 | |
| Market Value | 0.232381 | 0.037608 | 6.179069 | 0.0000 | Sig |
| Turnover | −10.79345 | 425.3501 | −0.025375 | 0.9799 | |
| Intercept | 22087.35 | 16537.6 | 1.335584 | 0.1928 | |

Source: authors.

The result of the above-fitted model is:
Equation (2):

$$R_{i+1} = 22{,}087.35 + 0.17\,E_i - 117.83\,F_i + 0.06\,C_i + 0.23\,V_i - 10.79\,T_i \tag{2}$$

From the illustration (Table 3) above, the column labelled "Prob" (*p*-values) records the two-tailed *p*-values for testing the hypothesis that each coefficient (Equation (2)) is different from zero, i.e.,:

$$\begin{cases} H_0 : \beta_i = 0 \\ H_1 : \beta_i \neq 0 \end{cases} \quad i = 1,\ 2,\ 3,\ 4,\ 5$$

To reject the above-stated null hypothesis, the *p*-value has to be lower than 0.05 (for 95% confidence). Alternatively, we also reject the null hypothesis when the *t*-statistic value is greater than 1.96.

The *p*-values also show the importance of a variable in the model. Thus, from our fitted model, the *p*-values show that the "number of employees" and "market value" variables have the most significant impact in predicting the revenues. This is because their *p*-values are equal to zero ($< 0.05$).

On the other hand, the other variables are not statistically significant since their *p*-values are greater than usual 0.05 (at 95% confidence). However, it is important to check the overall performance of our model. Table 6 below illustrates the model summary.

**Table 6.** Model summary.

| | |
|---|---|
| R-squared | 0.938217 |
| Adjusted R-squared | 0.926775 |
| F-statistic | 82.00213 |
| Prob (F-statistic) | 0.00000 |

Source: authors.

From the illustration above (Table 6), The model R-squared value is high (0.93). This indicates that our model explains 93% of the variance in our dependent variable (revenues M USD). Thus, our model fits the data quite well. It is important to examine the reliability of independent variables in predicting revenues M USD. This is done by observing the F-statistic. The *p*-value of the F-statistic is less than 1%. Thus, the goodness of fit is high which means that our model is good. This also indicates that our sample data provide sufficient evidence to conclude that our independent variables improve the model fit.

*4.2. Testing Homoscedasticity of Errors in Regression*

As the authors' assumption was that variance in the residuals was constant, a Breusch-Pagan-Godfrey test was used, which tests the heteroscedasticity in a linear regression. Thus, we tested the null hypothesis and the result was that the residuals are homoscedastic.

Table 7 shows the results of the Breusch-Pagan-Godfrey test for homoscedasticity. Given that the prob > chi-squared value of 0.0797 is greater than 0.05 (95% confidence) we fail to reject the null hypothesis at 95% and conclude that the residuals are homogeneous.

**Table 7.** Testing for homoscedasticity.

| Heteroscedasticity Test: Breusch-Pagan-Godfrey | | | |
|---|---|---|---|
| **Null Hypothesis: Homoscedasticity** | | | |
| F-statistic | 2.296334 | Prob. F (5,27) | 0.0732 |
| Observation R-squared | 9.846119 | Prob. Chi-Squared (5) | 0.0797 |

Source: authors.

## 5. Conclusions

The two variables of the number of employees and market value turned out to have the most significant impact in predicting revenues. This shows that well-educated employees and an experienced management team, including family members involved in the business, can influence strongly the development and growth of a company. Depending on the industry the company is in, another considerable aspect influencing the revenues is the number of employees available to manage the business when demand is increasing due to various reasons, such as a good economic situation or due to successful innovations. The other important factor in the results of the study is the market value of family businesses. Companies in a growing market or strongly developing industries (e.g., companies in online business like Amazon) have grown market values and this therefore positively impacts the revenue forecasts for the coming years. They are considered by private investors or private equity companies as interesting targets, their interests increase the market value and expectations are high for growing revenues [58]. All other variables in our empirical study are not statistically relevant (high % of family ownership over the shares; market capitalization), as the statistical calculations proved. With the performance check of the model, it showed very clearly that the R-squared value is high (0.95) for number of employees and market value when concerning revenues, which are of great importance to the statistical model. The right number of employees according to the business activity, the right management team, good managers with education and experience in the business

industry, and good communication between the employees and the management represent important factors that sustainable decisions impact the performance, here the revenues, in a very positive and controllable way. The other factor shown by the empirical study is the market value. Sustainable and successful decisions influence the value of the company, which is strongly influenced by the performance, the revenues, and EBITDA. With both mentioned variables (number of employees and market value), as shown in the tested empirical study, the revenue forecast for the next year can be anticipated. Regarding the weaknesses of the current research, the authors can investigate and identify for future research other factors which were not considered in this study, concerning revenues, such as heritage transfer in relation to stakeholders (employees, partners, unexperienced heirs) or national financial crisis and its effects for 5 years or effects within the hostile environment (dynamic industry effects over the family business) or even how international politics can affect or change laws and their impact on the revenues.

**Author Contributions:** All authors have equally contributed to the study and writing of this Article. In conclusion, all authors have the same rights on the paper. All authors have read and agreed to the published version of the manuscript.

**Funding:** This research received no external funding.

**Conflicts of Interest:** The authors declare no conflict of interest.

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
