# Peer review of "The Main Drivers for Sustainable Decisions in a Family Business That Impact the Company’s Performance"

_sustainability, doi:10.3390/su12208659_

Round 1
Reviewer 1 Report
Good, very good job.
Author Response
Thank you for your appreciation!
Reviewer 2 Report
The manuscript focuses on the main drivers behind sustainable decisions performed by family businesses. The topic is of interest, however, the manuscript has several weaknesses, as follows:
- The concept of sustainability is part of the title and presented in the abstract with reference to three pillars: economic, environmental and social. However, the work focuses on economic sustainability and the analysis is on the conditions that can ensure a medium-long term profitability of the company.
- Conclusions are very obvious and they add nothing new to the knowledge on the subject. In general, discussion and conclusions do not clarify how the study contributes to knowledge, nor what theoretical developments can derive from it. Furthermore, the managerial implications are not mentioned.
- The paper is written in a sometimes confusing way, with some passages not clear. Often the use of verbs is inconsistent, with present, past and future used without a uniform criterion. A linguistic review by a native speaker is required.
- The citations are made in a non-homogeneous way: in some cases authors are cited and the year reported in brackets instead of recalling the corresponding number of the references (eg page 4: "According to Zahra et al., (2008), the family business …………. According to Zellweger, (2017), culture ……. "), in other cases authors are cited without any reference to bibliography (eg page 3"… .many articles (Garcia-Castro; Chang; Rutherford, Mazzi ; Bases) and defined ……. "
- The numbering of the tables is confusing, the first two have no number, the third one is numbered as table 1, others are called inappropriately.
- The tables are formatted in a non-homogeneous way.
- The first table (not numbered) should report the sources of the main important drivers of the family business.
- The acronym COGS (page 8) is not explained.
- On page 7, regarding the first stage of the study, the employees are mentioned, but it is not explained how their value is taken into account.
- When the Breusch-Pagan-Godfrey test is mentioned, it is necessary to indicate bibliographic references for further information, since it is a not very well known test.
Given all of this, I believe there is still work to be done to make the manuscript acceptable.
Author Response
- Conclusions: The Authors have improved the conclusions by adding…xxxxxx……..and made also a discussion related to future developments of the results xxxxxx. In this context the Authors felt the need to better improve the Abstract by adding what the authors did in relation to the statistical model and on what our paper contribute for.
- Language: Authors have revised the Manuscript from language point of view
Answer: The Authors have done language modifications in the paperwork, where it was not clear understandable
- Citations:
Answer: The Authors improved the indicated citations (page 3, Garcia – Castro-Chang by transforming it into the corrects citation with the indicated year) and page 4, have clarified and modified the citation: “ page 3, e.g Barontini and Caprio, (2006) [16] diagnosed 675 family business from 11 countries, in 10 years, at the level of Europe. Sacristan-Navarro (2011) [17] has analyzed over 6 years in Spain that the family involvement is much more important than the ownership structure; Garcia-Ramos, (2011) [18] have concluded that the role of the family in the business influences the performance of the company.
- Counting of tables:
Answer: The Authors have chronologically counted the number of the Tables
- Source of the 1st table:
Answer: The Authors have added the source of the Table no 1-7, done by us, the Authors, which serves as contribution to the management family business literature
- Employees importance description:
Answer: Page 7 initially the employees characteristics have been explained, but the Authors better improved and clarified their connection with revenues. Even the employees represent an important cost position for the company, we consider them as added values for the companies because each employee contributes to the performance of the company by their experience and the fulfillment of a given task received from their managers in the company. As much as a family company has experienced and well-educated employees as much as it can take additional projects to grow the revenues. In this context, employees are considered an important asset for the company, considered cost – time efficient in their process activities and in strong connection with company’s growing revenues.
- The Breusch – Pagan- Godfrey test:
Answer: The Authors have better explained the meaning of the test in the context of the paperwork

Reviewer 3 Report
Dear authors,
I have several recommendations to your manuscript:
- Give main goal to the end of the Introduction and then set clearly description of the structure of the manuscript.
- Join Materials and Methods and Data and analysis to one understandable chapter.
- Define according to which references (authors) is your sample relevant.
- Add to Correlation matrix the significance of Correlation coefficients.
- The classic assumptions of used method is not only the Homoskedasticity. It is necessary to test all of them.
- Values of R-squared and Adjusted R-squared are high than 0.9. Do you really consider this model to be so good that it captures variability of more than 90%?
- There is no Discussion. Compare your results to other studies that dealt with this issue.
- What are the limits and weaknesses of your study?
- Set future development of solved issue.
- Do not use references in Conclusions.
I hope my comment will be useful for your future work.
Author Response
- Give main goal to the end of the Introduction and set clearly description of the structure of the manuscript: The Authors initially stated the goal of the Manuscript at the end of Introduction according to the last paragraph, but better clarified the objective of the paperwork in line with the Abstract (forecasting of the revenues). Additionally, the Authors add the study results, as well as weaknesses, as well as future framework of the implications research.
“To be able to forecast the future company revenue in the next financial year and to accomplish the objective of the paperwork, the Authors have deployed a descriptive analysis and an econometrical model on the top 100 family companies of the world”. The Authors present the study results, but also the weaknesses and create the framework for future research of practical family business implications. Our study results will contribute to the family business literature, by showing the important similitudes of the family businesses as well as their financial approach in doing sustainable decisions which impact on the future family business sustainable performance”
- Join Materials and Methods and Data and analysis to one understandable chapter: The Authors have split the Methods chapter into Sample, Procedure and Measures
- Add to Correlation matrix the significance of Correlation coefficients: The Authors added it to the title of the Table, also the Souces.
- The classic assumptions of used method are not only the Homoskedasticity. It is necessary to test all of them. The Authors didn’t find the need to test all of them since we found a good outcome with the R-square, the P-values, the F-statistic and the Chi-square.
- Values of R-squared and Adjusted R-squared are high than 0.9. Do you really consider this model to be so good that it captures variability of more than 90%? Yes, the Authors consider that the relation between the variables explains a 90% of the variation in the data. Especially when thinking that how many other variables could affect the revenue.
- There is no Discussion. Compare your results to other studies that dealt with this issue: At page 7 the Authors have included citations of other studies and in the Conclusion chapter the Authors referred to the future developments research:
“We have analyzed the surveys issued by three consulting companies from the Big Four group which have specialized departments for family business consultancy and conduct annual research on this issue, covering the period 2017–2018. Ernst & Young “How do we create value and build trust in this transformative age?” [41], PricewaterhouseCoopers (PwC) carried out Global Family Business Survey 2017/2018 [42]. The survey published by KPMG, which we used in our study, refers to the family businesses within the European Union (EU) [40, 43] and the report published by Deloitte “Connect for impact” [4]. The surveys mentioned above have been working on statistics for the family business which will be used in our equation”
- What are the limits and weaknesses of your study? The Authors had initially at the end of Conclusions the limitations included but have better clarified by adding: “All other variables in our empirical study are not statistically relevant, (high % of family ownership over the shares; market capitalization), as the statistical calculations proved. With the performance check of the model it showed very clear that the R-squared value is high, (0,95) for Number of employees and Market value in relation to Revenues, which gives to the statistical model a great importance”.
- Set future development of solved issue: Regarding the weaknesses of the current research, the Authors can investigate and identify for the future research other factors which were not considered in this study, in relation to revenues, such as: heritage transfer in relation with stakeholders (employees, partners, unexperienced hers) or national financial crisis and its effects for a 5 year period of time or effects within hostile environment (dynamic industry effects over the family business) or even how international politics can affect or change lows and its impact on the revenues
- Do not use references in Conclusions: The Authors deleted one citation (no. 51)

Reviewer 4 Report
Thank you for giving me a chance to examine a good paper.
It was a very interesting subject and research design was judged appropriate.
It is hoped that this research will have a positive effect on management decision making in practice.
Author Response
Thank you for your appreciation! Here attached, our Revision of the Manuscript.
Adriana Cioca

Round 2
Reviewer 2 Report
THE MANUSCRIPT HAS BEEN IMPROVED
Reviewer 3 Report
Dear authors,
thank you for your effort and comments. Your comment:
"The Authors didn’t find the need to test all of them."
Them = all classic assumptions of used method.
I have to correct the authors, the method used is not about the need of the authors, but clearly defined rules that they must follow in order to provide relevant results. If they do not tested the assumptions, what is the added value when the principles of method run are not implemented?
The achieved results are not valuable, if you did not test the classic assumptions.
That is why I have to recommend to reject your paper for scientific journal Sustainability.